# Doppler and Spectral Ultrasound of Sacroiliac Joints in Pediatric Patients with Suspected Juvenile Spondyloarthritis

**DOI:** 10.3390/diagnostics12040992

**Published:** 2022-04-14

**Authors:** Paolo Falsetti, Edoardo Conticini, Carla Gaggiano, Caterina Baldi, Maria Tarsia, Marco Bardelli, Stefano Gentileschi, Roberto D’Alessandro, Suhel Gabriele Al Khayyat, Alessandra Cartocci, Claudia Fabiani, Luca Cantarini, Maria Antonietta Mazzei, Bruno Frediani

**Affiliations:** 1Rheumatology Unit, Department of Medical Sciences, Surgery and Neurosciences, University of Siena, 53100 Siena, Italy; conticini.edoardo@gmail.com (E.C.); cgaggiano132@gmail.com (C.G.); catebaldi3@gmail.com (C.B.); mar.cobar@gmail.com (M.B.); gentileschi.stefano@gmail.com (S.G.); rt.dalessandro@gmail.com (R.D.); alkhayyatsuhelg@gmail.com (S.G.A.K.); cantarini@unisi.it (L.C.); fredianibruno60@gmail.com (B.F.); 2Clinical Pediatrics Unit, Department of Molecular Medicine and Development, University of Siena, 53100 Siena, Italy; tarsia.maria92@gmail.com; 3Department of Medical Biotechnologies, University of Siena, 53100 Siena, Italy; alessandra.cartocci@dbm.unisi.it; 4Ophthalmology Unit, Department of Medical Sciences, Surgery and Neurosciences, University of Siena, 53100 Siena, Italy; claudia.fabiani@gmail.com; 5Unit of Diagnostic Imaging, Department of Medical Sciences, Surgery and Neurosciences and of Radiological Sciences, University of Siena, 53100 Siena, Italy; mamazzei@gmail.com

**Keywords:** juvenile spondyloarthritis, pediatric, sacroiliitis, power doppler ultrasound, magnetic resonance imaging

## Abstract

Background: Power Doppler ultrasound (PDUS) with spectral wave analysis (SWA) has been compared with magnetic resonance imaging (MRI) in documenting active sacroiliitis in early spondyloarthritis (SpA) but, to date, PDUS/SWA has not been yet applied to the study of sacroiliac joints (SIJs) in children. Methods: A group of 20 children (13 F/7 M, mean age 14.2 y) with suspected juvenile SpA (jSpA) underwent PDUS/SWA and, subsequently, MRI of the SIJs. SIJs PDUS scoring and resistance index (RI) of the SIJs flows were recorded. The accuracy of PDUS/SWA for the diagnosis of active sacroiliitis was evaluated, with MRI as the gold standard. Results: PDUS signals were detected in 19 patients and 30 SIJs. Bone marrow edema (BME) lesions on MRI were detected in 12 patients (diagnosed as jSpA) and 22 SIJs. PDUS scoring on SIJs were higher in patients with a final diagnosis of jSpA (*p* = 0.003). On SWA, the mean RIs in patients with or without final diagnosis of active sacroiliitis were, respectively, 0.604 and 0.767 (*p* = 0.005) at joint level. A RI < 0.55 and PDUS > 1 showed the higher specificity for sacroiliitis (AUROC curve 0.854 for PDUS and 0.920 for RI). SIJs PDUS/SWA showed an overall concordance of 82.35%, with substantial agreement (k = 0.627) with MRI on the diagnosis of sacroiliitis. Conclusions: In children with sacroiliitis, PDUS demonstrates a rich vascularization into SIJs and low RIs (<0.55) have high specificity for this condition. SIJs PDUS/SWA could be useful as a screening method in children with suspected jSpA.

## 1. Introduction

Imaging plays a pivotal role in the assessment of suspected sacroiliitis in children and magnetic resonance imaging (MRI) is the gold standard for the diagnosis of early disease. MRI findings consistent with sacroiliitis greatly influence treatment decisions for children, even if in contrast to adult spondyloarthritis (SpA)—where the Assessment of SpA International Society (ASAS) diagnostic criteria include MRI findings—imaging has no role in the current International League of Associations for Rheumatology (ILAR) classification criteria for Juvenile Idiopathic Arthritis (JIA) [1]. Nevertheless, recent studies have suggested the usefulness of MRI in juvenile SpA (JSpA) [2].

The routine application of MRI in children with suspected sacroiliitis can encounter some difficulties as variability in MRI interpretation of pediatric sacroiliac joints (SIJs) due to the changes that occur with skeletal maturation (metaphyseal equivalent signal intensity and red bone marrow spots mimicking bone marrow edema) and the necessity of sedation for younger children, with dedicated MRI session [3,4]. Moreover, the role of contrast-enhanced sequences and the applicability of ASAS criteria in pediatric sacroiliitis are still debated [5,6]. Finally, from a clinical point of view, pediatric rheumatologist can encounter difficulties on suspecting sacroiliitis in children, as younger patients do not report clear symptoms of inflammatory back pain. In this context, pediatric rheumatologists could benefit from a not-invasive, quick, and cheap examination to support their MRI prescription.

In a few studies, ultrasound (US) and Doppler US with spectral wave analysis (SWA) were used as first assessment tool to detect active sacroiliitis in adult patients [7,8,9,10,11,12,13,14,15]. More recently, Ghosh et al. and Falsetti et al. suggested that Power Doppler US (PDUS) with SWA may be a non-inferior diagnostic modality compared with MRI in documenting active sacroiliitis in early SpA [16,17]. On the other hand, Doppler US has not been yet applied to the study of SIJs in children [18,19], and The European Society of Musculoskeletal Radiology (ESSR) arthritis subcommittee and the European Society of Paediatric Radiology (ESPR) musculoskeletal imaging taskforce attributed limited value to the usefulness of US in axial joints [20].

In this context, we investigated the use of unenhanced SIJs PDUS and SWA in order to detect the presence of active sacroiliitis in a cohort of pediatric outpatients with suspected JSpA before performing a diagnostic SIJs MRI, in a scenario of real clinical practice.

## 2. Methods

### 2.1. Study Sample

Twenty consecutive outpatients with a suspicion of jSpA, assessed by a pediatric rheumatologist, were included in the study. Patients were only included when age at onset of disease was <16 years. All the patients were recruited in a single third level rheumatology center between March 2020 and April 2021. Patients were addressed to US and MRI study of the SIJs if a clinical suspicion of jSpA was formulated according to: (1) a history of new-onset inflammatory back pain according to the ASAS criteria, (2) a history of any lumbar pain in a child with JIA, (3) the presence of SIJs tenderness at the physical examination, (4) the suspect of arthritis in a child with a personal or first grade familial history of any HLA B27-related conditions. The following clinical and laboratory data were collected at baseline: anthropometric data including age and BMI, history of the disease, main articular (past or current enthesitis, dactylitis, arthritis) and extra-articular symptoms (past or current uveitis, psoriasis, inflammatory bowel disease), inflammatory markers such as CRP and ESR, presence/absence of HLA-B27, pharmacological treatments. The composite disease activity score (juvenileSpA Disease Activity, jSpADA) was calculated for patients affected by jSpA [21].

The SIJs of 20 asymptomatic healthy children (sons and daughters of the hospital staff) were evaluated with PDUS/SWA to define normative values.

The study protocol conformed to the tenets of the Declaration of Helsinki and was approved by the local Ethics Committee of the University of Siena (Reference No. 14951). Written informed consent for using clinical data for research purposes was obtained according to the local Institutional review board guidelines. Informed consent was also obtained from legal guardians, where applicable, for all the procedures.

### 2.2. Procedures and Measurements

All patients underwent US and subsequent MRI of the SIJs within two weeks, before starting any pharmacologic treatment. In order to study the SIJs, every patient underwent a US examination in prone position. The sonographer was blinded to the clinical data of the patient, except for the indication to the examination. A Esaote (Genova, Italy) MyLab Twice US machine, equipped with a convex multifrequency 1–8 MHz probe (CA541) was used, with standardized parameters (factory preset of the machine for musculoskeletal): 1–8 MHz for greyscale, 1.9–2.3 MHz frequency for Power Doppler with Pulse Repetition Frequency (PRF) of 0.750–1.3 KHz, and a color gain just under the artefact limit; 1.9–2.1 MHz frequency for pulsed wave (PW) Doppler with a pulse repetition frequency (PRF) of 0.750–2.5 KHz.

The US examination of SIJs was performed as described in previous papers [8,9,10,11,12,13,14,15,16,17] (Figure 1A). The probe was positioned in transverse scan over the median line of the sacrum, at the spinous apophysis of first sacral vertebra. The probe was then shifted over the top of the left first sacral foramen up to the margin of the ilium, where the probe was placed in an oblique position and the sound beam inclined about 20° downwards. The process was then repeated ab initio for the right SIJ. The SIJ appeared as a hypoechoic triangle delimited between the sacrum and iliac bone, with the posterior SI ligament as the upper margin. The first sacral foramen was always observed to avoid measurement of the pre-sacral arteries and veins. All extensions of the SIJs were explored up until the second sacral foramen level. Unlike examination in adult patients, the appearance of first sacral spinous apophysis appeared more hypoechoic because of the presence of epiphyseal cartilages.

PDUS was applied, starting from the left SIJ and then shifting to the right side. The color box was reduced to avoid artifacts and improve the sensitivity of Doppler, but it was extended at least from the iliac border to the sacral foramen (Figure 1B). The first sacral foramen was always observed to verify the regular presence and spectral pattern of pre-sacral arteries and veins (Figure 1C).

The posterior–superior portion of SIJ (near first sacral foramen) shows the more abundant vascularization originating from posterior division of superior gluteal artery (often an arterial vessel along the posterior sacroiliac ligament and another archiform vessel penetrating the SIJ, with negative spectrogram with respect to foraminal flows). In this portion, the interosseus ligaments are more evident, with possible vascularization. At the more caudal level (near second sacral foramen) SIJ is tight, with a superficial vascularization derived from branches of anterior division of superior gluteal artery (almost always with high RI). The measurements are performed in the posterior–superior portion of SIJs, with peculiar attention to interosseous flows (often with positive spectrogram, as these vessels originate from deeper portion of SIJ).

If any power Doppler signal was detected in the SIJ space, it was scored using a 3-point scale: 0 = absence of signal, 1 = isolated vessel, 2 = more than one vessel. The same vessels were also evaluated using quantitative PW Doppler and SWA, manually calculating the Resistive Index (RI = peak of systolic flow − end diastolic flow/peak systolic flow) (Figure 1D). At least two measurements were obtained for each vessel, and the mean value was recorded. Only intra-articular arterial pulsatile flows were computed.

Diagnosis of positive PDUS/SWA examination for active sacroiliitis was defined as: PDUS score > 0 and RI < 0.60, or PDUS score = 2 and RI < 0.70 [17].

All the patients were also evaluated with routine multi-site bilateral dynamic B-mode and PDUS entheseal and joint examination (at least elbows, knees, and feet) using a linear multifrequency 6–18 MHz probe (LA435) and standardised B-mode and Doppler settings. US diagnosis of enthesitis and synovitis were based on OMERACT’s definitions [22,23,24]. Belgrade Ultrasound Enthesitis Score (BUSES) [25] and global US count of enthesitis and synovitis were computed for each patient [26].

All MRIs were performed with a 1.5 T MR scanner (Signa Twin Speed Hdxt; GE Healthcare, Milwaukee, WI, USA) by the same radiologist, trained in axial SpA diagnosis and belonging to the Radiology unit of the same hospital. The radiologist was aware of the suspected diagnosis, but blinded to the US results. The diagnostic reports for the MRI were based on ASAS criteria [2,6,20]: active sacroiliitis was defined as the presence of SIJ bone marrow edema (BME) lesions on short-tau inversion recovery images (STIR) [2,6,27,28]. Spinal involvement was reported only for lower lumbar vertebral levels comprised in the study [29]. The MRI could be completed with i.v. contrast media administration (if it was considered necessary by the radiologist and if parents gave their written informed consent to use of contrast).

### 2.3. Statistical Analysis

Data are reported as mean and standard deviations (SD) for continuous variables, whereas categorical and dichotomous variables are reported as frequencies and percentages. Student’s *t*-test was used to compare the means of continuous variables between two groups when the distribution of data was normal, and with Welch’s correction otherwise.

The non-parametric Kruskal–Wallis testing was used to compare the means of continuous variables among groups, with Dwass–Steel–Critchlow–Fligner (DSCF) test for pairwise comparisons.

Fisher’s exact test was used to compare the percentages between two groups for categorical variables. The non-parametric Spearman rank test was applied in order to correlate variables. The tests regarding US diagnostic properties were applied both at patient level (considering only the most pathologic value for each patient, regardless of the side) and at joint level (considering the values obtained for the same joints).

Cohen’s k statistics were used to assess the agreement between the two alternative methods of categorical assessment (PDUS/SWA and MRI diagnosis of sacroiliitis).

Moreover, binomial logistic regression and receiver operating characteristic (ROC) curve analysis were used to determine predictive diagnostic value of each parameter of PDUS/SWA detected sacroiliitis with MRI as gold standard. Validity of each parameter of PDUS/SWA for diagnosis of active sacroiliitis was determined by the estimation of sensitivity and specificity of various cut-off points of both PDUS grading and of RI values. Youden’s J statistic and closest-topleft methods were applied to obtain the optimal cut-off values.

The level of statistical significance was set at a *p*-level of 0.05. Statistical analyses were performed using Jamovi and RStudio statistical packages. Feasibility of SIJs PDUS/SWA was evaluated by recording the time spent by the operator and asking the patient and its parents about the comfortability of the examination.

## 3. Results

Twenty patients (13 females and 7 males, mean age 14.2 ± SD 2.95, range 7–17 years) and 40 SIJs were studied. The demographic, anthropometric and clinical characteristics of the patients are reported in Table 1.

In 20 healthy children (8 females and 12 males, mean age 9.4 years ± 3.6, range 5–14 years), PDUS signals were detected in all the subjects and in 30 SIJs, with a mean PDUS score of 0.75 (±0.439, range 0–1) and a mean RI of 0.73 (±0.161, range 0.58–1). The mean coefficient of variation for repeated measurements of RI in static images of the same joint was 3.81%.

Power Doppler signals were detected in 19 patients and in 30 SIJs. BME lesions on MRI were detected in 12 patients and in 22 SIJs. Nine SIJs showed vascularization in PD, with no BME lesion on MRI.

A final diagnosis of jSpA was made in 12 patients (all with at least SIJs BME on MRI, and five patients with also spinal inflammatory involvement). Among the jSpA patients, three had enthesitis, six had synovitis, two had skin psoriasis, one had uveitis, six were B27+. The jSpA patients does not showed significant differences about mean BMI percentile, age, ESR and PCR with respect to not-jSpA.

Patients with a final diagnosis of jSpA had a significantly higher PDUS score (1.13, *p* = 0.003) at joint level, when compared with not-jSpA (0.56). A correlation was demonstrated between PDUS score and presence of BME in MRI at patient level (r = 0.519, *p* = 0.019).

For SWA on Doppler US, the mean RIs in patients with or without active sacroiliitis were, respectively, 0.604 and 0.767 (*p* = 0.005) at joint level, and 0.534 and 0.656 (*p* = 0.024) at patient level. An inverse correlation was demonstrated between RI and presence of BME in MRI at patient level (r = −0.586, *p* = 0.007) (Figure 2).

In healthy subjects, PD scores (0.75 ± 0.439, range 0–1) and RIs (0.736 ± 0.161, range 0.58–1) at joint level were significantly different than those values obtained in jSpA patients (*p* = 0.026 and *p* = 0.001 respectively), but not significantly different with respect to not-jSpA patients (*p* = 0.359 and *p* = 0.969 respectively) (Figure 3).

No correlation was obtained between PDUS/SWA diagnosis and either age, BUSES, inflammatory reactants, enthesitis, and synovitis count.

The Cohen statistic between the two diagnostic methods showed an overall concordance of 82.35%, with substantial agreement (k = 0.627), with standardized cut-off for PDUS diagnosis.

A ROC curve was calculated considering each Doppler parameter (PDUS grading and SWA-RI values) in the diagnosis of active sacroiilitis, in comparison with the presence of BME lesions on MRI as the gold standard.

For PDUS grading, an estimated area under the curve (AUC) of ROC curve of 0.854 was found, with a sensitivity of 1, and a specificity of 0.562 with cut-off PD score of 1, and a sensitivity of 0.33, and a specificity of 1 with cut-off PD score of 2.

For RI values an estimated AUC of ROC curve of 0.920 was found, with a sensitivity of 0.778, and a specificity of 0.875 with cut-off RI of 0.595 (closest-topleft), and a sensitivity of 0.722, and a specificity of 1 with cut-off RI of 0.55 (Youden).

Using the maximal specificity of Doppler cut-offs for diagnosis of active sacroiliitis (PD score = 2 and RI < 0.55), the Cohen statistic between the two diagnostic method showed a lower overall concordance of 64.7%, with fair agreement (k = 0.354).

A time of 2–3 min was sufficient for setting the US parameters, and for completing the greyscale and PDUS examination on both SJJs, whereas a time of 3–4 min could be necessary to complete the SWA and RI measurement on both SIJs. No adverse events occurred during examinations, and all patients and their parents considered this examination quick, not painful, and mostly comfortable.

## 4. Discussion

A few previous publications have suggested a possible role of Doppler US in determining active sacroiliitis in adults [8,9,10,11,12,13,14,15,16,17,18,19], demonstrating a high specificity of this examination, in comparison with MRI as gold standard, and proposing it as a screening method for patients with inflammatory back pain.

To date, this is the first study in which PDUS and SWA are applied to the study of SIJs in pediatric patients with suspected jSpA, in a context of a real clinical scenario.

The application of MRI in children with suspected sacroiliitis remains the gold standard for evaluation of early disease, but it can encounter some difficulties in clinical routine because of the necessity of sedation for younger children, long waiting lists for dedicated sessions, and variability in MRI interpretation of pediatric SIJs due to the changes that occur with skeletal maturation [3,4,6].

We have therefore applied PDUS and SWA in a small case series of children with suspected sacroiliitis, testing this tool as a screening procedure before MRI, using previous US criteria obtained for adults [17].

Our study confirms that, also in pediatric patients, SIJs PDUS examination is a quick and comfortable procedure, with a substantial agreement with MRI diagnosis of active sacroiliitis, using previously published criteria [17].

PDUS signals into SIJs resulted more frequent and intense in children with final diagnosis of jSpA, but conversely from adult patients, in most of non-pathologic SIJs, PDUS signals could be commonly observed, although of lower scores than in patients with sacroiliitis. This aspect can explain why the AUROC of PD score is lower than AUROC-RI. Moreover, in comparison with our previous experience in SIJs PDUS of adults, we can observe a substantially better visualization of sacroiliac structures in younger patients, because of absence of fibrosis and calcification of ligamentous structures occurring in adults. The more prominent vascularization in pediatric joints has already been underlined in previous works and it should be taken into account also for the PDUS study of pediatric SIJs [23,30]. However, the SWA showed RI cut-off values slightly lower than the adults’ ones, so this analysis cannot be separated from the PDUS semi-quantitative scoring.

Furthermore, in our case series, we experienced difficulties in applying MRI in at least one pediatric patient, in which the radiologic interpretation of MRI gave not conclusive results in a single SIJ about a hyperintense spotty area (possibly red marrow island) not evaluable with contrast enhancement because of lack of consent from parents. In cases like this, we tried to use more specific PDUS criteria to assess active sacroiliitis, aiming to use PDUS with diagnostic purposes. Therefore, we have re-tested each PDUS/SWA parameter, calculating AUROC for both PD score and RI values, choosing the more specific cut-off to guarantee a higher specificity of the examination (at the expense of a reduction in sensitivity). Using the maximal specificity of Doppler cut-offs for the diagnosis of active sacroiliitis (PD score = 2 and RI < 0.55), the concordance with MRI diagnosis fell to a fair agreement. Nevertheless, in the aforementioned case (overweight children with Osgood–Schlatter apophysitis and first-grade familiarity for psoriasis), the standardized PDUS criteria gave suggestion of active sacroiliitis (PDUS score 1 and RI 0.58), but using the most specific cut-offs (PDUS score = 2 and RI < 0.55) this patient could be classified as not pathologic about SIJs (in the present study, this child had been finally classified as non-jSpA and as a false positive for PDUS analysis).

Further studies on this topic should be guaranteed, also testing the performance of new technologies—such as microvascular imaging software (i.e., Micro-v for Esaote)—and using linear probes with better spatial resolution. In particular, linear probes could have a role in the study of pediatric SIJs, because of the reduced thickness of subcutaneous fat and favorable anthropometric measures. The better resolution of linear probes could be useful in the chronic stages of jSpA to assess enthesopathic changes and thickness of posterior sacroiliac ligament, which are described in degenerative SIJs diseases [31,32]. Contrast-enhanced US (CEUS) has been used in adult patients, demonstrating higher sensibility for sacroiliitis [9]. However, this diagnostic tool could encounter several difficulties in children (invasiveness, movement artifacts, short tolerance to long examination in younger patients).

Some limitations of the present study should be discussed. The main limitation of our study is the low sample size, which reduces the strength of our findings: further studies should include a larger number of patients, from different centers and with standardized US procedures.

Secondly, our study lacks a complete control group, represented by healthy patients, which is difficult to overcome given the unethical requirement of unnecessary invasive exams (i.e., procedural sedation or contrast MRI) in healthy children. However, we reported normative SIJs PDUS/SWA evaluation of a small series of pediatric healthy subjects.

Lastly, differently than how we proceeded with adult patients in our previous study [17], only one sonographer performed US to minimize discomfort for the children, making it impossible to calculate inter-observer agreement. However, we reported the CV of repeated measurements of RI as indicator of intra-observer reliability.

In conclusion, PDUS can detect active sacroiliitis, with significant correlation with MRI findings, also in pediatric patients. The high specificity of PDUS/SWA examination, in particular of SWA of SIJs flows, makes it possible to use this tool as a screening method in pediatric patients. Further studies should be designed to assess the reliability of US exam as a diagnostic tool to support MRI imaging, when this is not conclusive or not easily available, PDUS/SWA criteria should be used with the highest specificity.

## Figures and Tables

**Figure 1 diagnostics-12-00992-f001:**
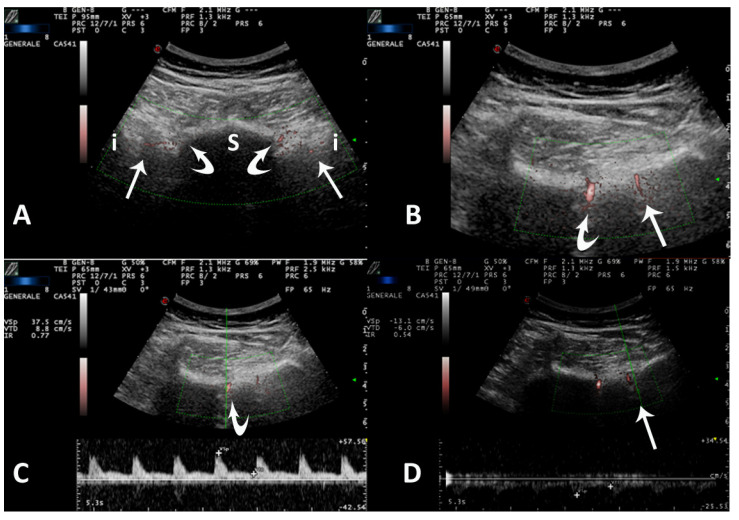
Axial posterior scan over SIJs in a 15−year−old girl with jSpA. (**A**) Panoramic view that includes both SIJs (straight arrows). The profile of sacrum (s) is interrupted by the first sacral foramen (curved arrow) on both sides. i = ilium. Vascular signals can be observed into both sacral foramen and SIJs. (**B**) In a more distal axial scan over right side, clear vascular flows can be observed in the sacral foramen (curved arrow) and in the cleft of right SIJ (straight arrow). (**C**) The application of pulsed wave (PW) Doppler with spectral wave analysis (SWA) on the signal of sacral foramen (curved arrow) shows a high resistance normal flow (RI 0.77). (**D**) The application of pulsed wave (PW) Doppler with spectral wave analysis (SWA) on the signal of sacroiliac cleft (straight arrow) shows a low resistance flow (RI 0.54) suggesting vasodilatation due to SIJ inflammation.

**Figure 2 diagnostics-12-00992-f002:**
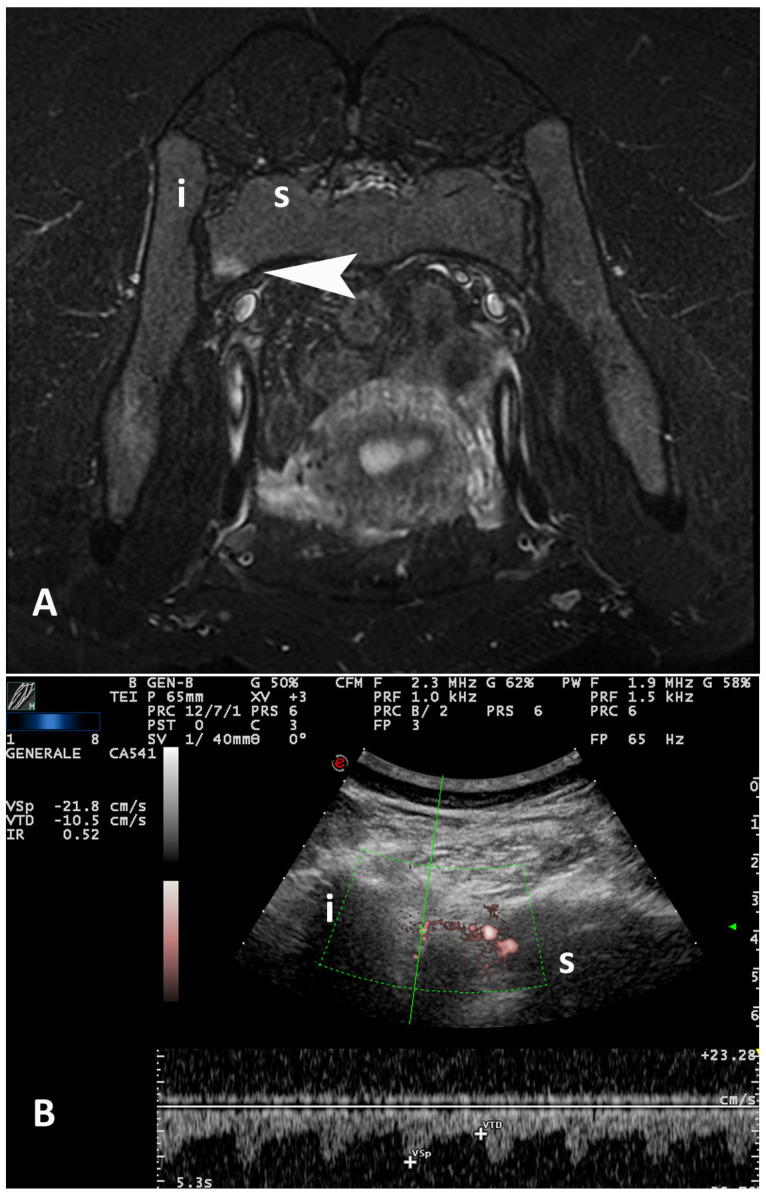
17−year−old girl with jSpA. (**A**) Magnetic resonance imaging (MRI) (the image has been rotated of 180° for a better comparison with the corresponding ultrasound scan): a short−tau inversion recovery (STIR) axial image shows a focal hyper−intense area of bone marrow edema (arrowhead) in the sacral side of left SIJ due to sacroiliitis. (**B**) PDUS/SWA over the same SIJ demonstrates a low resistance flow (RI 0.52), suggesting vasodilatation due to sacroiliitis. The majority of vessels observed in the posterior−superior portion of SIJ originate from branches of the posterior division of superior gluteal artery, which emerge from sacral foramina (note the intense vascular signals into the sacral foramen). i = ilium; s = sacrum.

**Figure 3 diagnostics-12-00992-f003:**
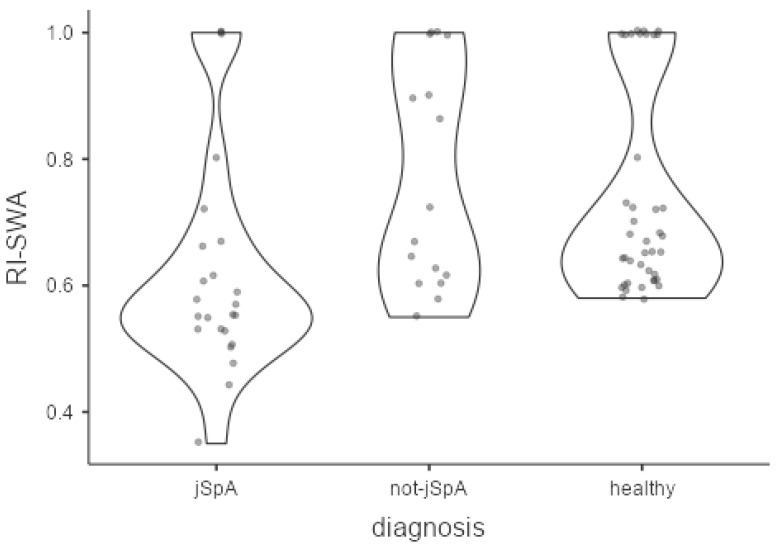
Comparisons of RI-SWA among the groups expressed as violin plots. The Kruskal–Wallis test showed significantly differences in RI values among groups (*p* < 0.001). The pairwise comparisons demonstrated lower RIs in jSpA respect both to not-jSpA patients (*p* = 0.005) and healthy subjects (*p* = 0.001), but not significant difference between not-jSpA patients and healthy subjects (*p* = 0.969).

**Table 1 diagnostics-12-00992-t001:** Demographic, anthropometric, and clinical characteristics of patients. Data are expressed as mean (±standard deviation, SD), if not otherwise specified. The level of statistical significance was set at a *p*-level of 0.05. n.s. = not significant; * = *p* < 0.05; ** = *p* < 0.01; n.a. = not assessed. IBP = inflammatory back pain; jSpA = juvenile spondyloarthritis; PDUS = power Doppler ultrasound; BMI = body mass index; ESR = erytrosedimentation rate; CRP = C-reactive protein; JSpADA = juvenile spondyloarthritis disease activity; SIJ = sacroiliac joint; RI = resistive index; BME = bone marrow edema; BUSES = Belgrade Ultrasound Enthesitis Score.

Patients Characteristics	Overall Population	Final jSpA Diagnosis	Not jSpA	Statistical Significance
Number of patients (female/male)	20 (13/7)	12 (8/4)	8 (5/3)	n.s.
Age, in years (±SD)	14.2 (±2.95)	14.6 (±2.75)	13.9 (±2.53)	n.s.
BMI percentile(±SD)	67.6 (±26.8)	74.3 (±24)	54.2 (±29.7)	n.s.
ESR mm/h(±SD)	23.8 (±22.9)	28.9 (±25.2)	11 (±8.08)	n.s.
CRP mg/dL (±SD)	1.04 (±1.58)	1.31 (±1.79)	0.51 (±1.01)	n.s.
jSpADA (±SD)	2.75 (±1.89)	2.75 (±1.89)	n.a.	n.a.
-Patients with PDUS+ -SIJs with PDUS+	19 patients (95%) 30 SIJs (75%)	12 patients (100%) 21 SIJs (87.5%)	7 patients (87.5%) 9 SIJs (56.2%)	n.s. n.s.
SIJs PD grading (±SD) patient level	1.20 (±0.523)	1.42 (±0.515)	0.875 (±0.354)	*p* = 0.012 *
SIJs PD grading (±SD) joint level	0.912 (±0.668)	1.13 (±0.612)	0.563 (±0.512)	*p* = 0.003 **
SIJs RI (±SD) patient level	0.583 (±0.115)	0.534 (±0.092)	0.656 (±0.112)	*p* = 0.024 *
SIJs RI (±SD) joint level	0.671 (±0.181)	0.604 (±0.155)	0.767 (±0.176)	*p* = 0.005 **
MRI SIJs BME lesions	12 patients (60%) 22 SIJs (55%)	12 patients (100%) 22 SIJs (91.6%)	0 0	n.a.
MRI SIJs capsulitis/enthesitis	6 patients (30%) 11 SIJs (27.5%)	6 patients (50%) 11 SIJs (45.8%)	0 0	n.a.
MRI lumbar spine inflammatory involvement	5 patients (25%)	5 patients (41.6%)	0	n.a.
Patients with US enthesitis total amount of US enthesitis BUSES (mean)	4 patients (20%) 8 1.8	3 (25%) 6 1.83	1 (12.5%) 2 1.75	n.s. n.s. n.s.

## Data Availability

Not applicable.

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
