# Peer review of "Doppler and Spectral Ultrasound of Sacroiliac Joints in Pediatric Patients with Suspected Juvenile Spondyloarthritis"

_diagnostics, 2022, doi:10.3390/diagnostics12040992_

Round 1

Reviewer 1 Report

The authors presented an interesting study, however, several problems appeared. 

  1. Key message section should be a part of the abstract and not the separate section. 
  2. Introduction does not give a holistic view on the problem. How do children joints differ with adult ones in case of MRI use? Why using ultrasounds instead of MRI?
  3. Line 87 - missing space bar " was<16"
  4. Figure 1 and 2 are included in the materials and methods section - maybe it would be better to shift them to results?
  5. The sample size is small but the authors can support the data with wider discussion of other authors' studies... Otherwise a nice paper! 

Author Response

Please see the attached file for the answers to Your observations.

Reviewer 2 Report

Using PDUS is promising although 82% reported by authors doesn't look accurate enough to replace MRI particularly for children.

More case studies at different age and at different stages of disease need to be studies to generalize the findings of this study.

One main concern is that in the discussion authors wrote that "PDUS can detect active sacroiliitis". What about non-active ones? MRI is able of capturing non-active ones so do authors take it as limitation for PDUS? Have authors examined non-active ones? 

Author Response

(The authors gave the same response as above.)

Reviewer 3 Report

Review on Doppler and spectral ultrasound of sacroiliac joints in pediatric patients with suspected juvenile spondyloarthritis

The paper is well written and interesting.

Major concerns:
It seems that the correlation is relatively weak between your findings with ultrasound and MRI. Even though some of the p-values are significant, the major concern is how reliable is the tool in practice. You need two different ultrasound parameters in combination as neither PDUS nor SWA can’t stand alone. The combination of PDUS/SWA overestimate juvenile spondyloarthritis compared to MRI. Looking at Fig 3 it is evident that there is a substantial overlap of SWA-values between the groups.

Should the method be used to exclude pts or used to find pts for the more reliable follow-up MRI exam for final diagnosing of juvenile spondyloarthritis. You write in the discussion that parameters are set to a high specificity, so I reckon you seek to exclude the healthy. How many in your population of 20 pts with symptoms would have been excluded for the following MRI? Would you have had any false negative cases with your cut-off?

I would like a discussion of the results, where and how to use US in diagnosing juvenile spondyloarthritis. Also, some suggestions of how to improve the performance of the method as conventional Doppler is an old technique – could suggestions be microflow with super-resolution imaging, CEUS or AI?

Minor concerns:

It would be beneficial if a photo/illustration of placement of the probe with an anatomical drawing combined with a US image for better illustration of the scan technique as the SI-joint can be difficult to assess.

You write the all extension of the SIJ were explored. How did you decide where to measure? Whenever you found a vessel? At which level of the SIJ?

I wonder if the flow in the SIJ always is negative in SIJ compared to the positive flow in the foramen? Could this be used as a marker?

In fig 1D it is difficult to see the spectrogram – do you have a better example?

Could you illustrate the PDUS score eg. in a boxplot for the different groups like you did for SWA in fig 3.

Could you in table 1 give number of samples (n) for each comparison.

I’m a bit puzzled by the extra parameters given in the paper – ex capsulitis, enthesitis etc. As I see the paper, it is a comparison between MRI and US of the SIJ. Do these extra parameters change the results of the paper in regard to the correlation between MRI and US? If the SIJ is normal in MRI, it should be normal with US?

Please explain the use of the findings of other joints and lumbar spine inflammation in relation to the correlation or exclude them.

Author Response

(The authors gave the same response as above.)

Round 2

Reviewer 1 Report

The authors heavily revised the paper according to my comments and now the paper may be considered for publication in Diagnostics.